# Mismatch Repair Deficiency Is a Prognostic Factor Predicting Good Survival of *Opisthorchis viverrini*-Associated Cholangiocarcinoma at Early Cancer Stage

**DOI:** 10.3390/cancers15194831

**Published:** 2023-10-02

**Authors:** Natcha Khuntikeo, Sureerat Padthaisong, Watcharin Loilome, Poramate Klanrit, Soontaree Ratchatapusit, Anchalee Techasen, Apiwat Jareanrat, Vasin Thanasukarn, Tharatip Srisuk, Vor Luvira, Jarin Chindaprasirt, Prakasit Sa-ngiamwibool, Chaiwat Aphivatanasiri, Piyapharom Intarawichian, Supinda Koonmee, Piya Prajumwongs, Attapol Titapun

**Affiliations:** 1Department of Surgery, Faculty of Medicine, Khon Kaen University, Khon Kaen 40002, Thailand; natcha766@gmail.com (N.K.); apiwat.atj@kku.ac.th (A.J.); vasith@kku.ac.th (V.T.); tharsr@kku.ac.th (T.S.); vor@kku.ac.th (V.L.); 2Cholangiocarcinoma Research Institute, Khon Kaen University, Khon Kaen 40002, Thailand; watclo@kku.ac.th (W.L.); porakl@kku.ac.th (P.K.); sr.ratchatapusit@gmail.com (S.R.); anchte@kku.ac.th (A.T.); jarich@kku.ac.th (J.C.); prakasa@kku.ac.th (P.S.-n.); chaiap@kku.ac.th (C.A.); piyain@kku.ac.th (P.I.); suppun@kku.ac.th (S.K.); prajumwongspiya@gmail.com (P.P.); 3Faculty of Allied Health Sciences, Burapha University, Chonburi 20131, Thailand; sureerat.pa@go.buu.ac.th; 4Systems Biosciences and Computational Medicine, Faculty of Medicine, Khon Kaen University, Khon Kaen 40002, Thailand; 5Faculty of Associated Medical Sciences, Khon Kaen University, Khon Kaen 40002, Thailand; 6Medical Oncology Program, Department of Medicine Srinagarind Hospital, Khon Kaen University, Khon Kaen 40002, Thailand; 7Department of Pathology, Faculty of Medicine, Khon Kaen University, Khon Kaen 40002, Thailand

**Keywords:** cholangiocarcinoma, *Opisthorchis viverrini*, mismatch repair proteins, microsatellite instability, prognosis, early stage, chemotherapy

## Abstract

**Simple Summary:**

The mismatch repair (MMR) system prevents DNA mutations, and deficient MMR protein (dMMR) can lead to genetic changes and microsatellite instability (MSI). While dMMR is typically associated with positive outcomes in various cancers, its role in cholangiocarcinoma (CCA) remains uncertain. This study’s objective was to assess the prevalence of dMMR in CCA patients and examine its relationship with clinicopathological features and patient survival following surgery. This study showed that dMMR was present in 22.5% of CCA patients and was associated with better survival, especially in early-stage CCA and when combined with adjuvant chemotherapy. This study suggests that dMMR could be a valuable marker for selecting CCA patients for specific adjuvant treatments after surgery.

**Abstract:**

Background: The mismatch repair (MMR) system prevents DNA mutation; therefore, deficient MMR protein (dMMR) expression causes genetic alterations and microsatellite instability (MSI). dMMR is correlated with a good outcome and treatment response in various cancers; however, the situation remains ambiguous in cholangiocarcinoma (CCA). This study aims to evaluate the prevalence of dMMR and investigate the correlation with clinicopathological features and the survival of CCA patients after resection. Materials and Methods: Serum and tissues were collected from CCA patients who underwent resection from January 2005 to December 2017. Serum OV IgG was examined using ELISA. The expression of MMR proteins MLH1, MSH2, MSH6 and PMS2 was investigated by immunohistochemistry; subsequently, MMR assessment was evaluated as either proficient or as deficient by pathologists. The clinicopathological features and MMR status were compared using the Chi-square test. Univariate and multivariate analyses were conducted to identify prognostic factors. Results: Among the 102 CCA patients, dMMR was detected in 22.5%. Survival analysis revealed that dMMR patients had better survival than pMMR (HR = 0.50, *p* = 0.008). In multivariate analysis, dMMR was an independent factor for a good prognosis in CCA patients (HR = 0.58, *p* = 0.041), especially at an early stage (HR = 0.18, *p* = 0.027). Moreover, subgroup analysis showed dMMR patients who received adjuvant chemotherapy had better survival than surgery alone (HR = 0.28, *p* = 0.012). Conclusion: This study showed a high prevalence of dMMR in cholangiocarcinoma with dMMR being the independent prognostic factor for good survival, especially in early-stage CCA and for patients who received adjuvant chemotherapy. dMMR should be the marker for selecting patients to receive a specific adjuvant treatment after resection for CCA.

## 1. Introduction

Cholangiocarcinoma (CCA) is a tumor arising from bile duct epithelium. CCA has a relatively rare incidence in most Western countries and North America [1,2,3]. However, it has its highest incidence in southeast Asia especially in the northeast of Thailand with 135.4 per 100,000 among males and 43.0 per 100,000 among women [1,4]. Surgery is the only curative treatment for CCA patients. Owing to the asymptomatic characteristics of CCA, most patients have advanced disease due to late diagnosis, which causes poor prognosis and outcomes. The five-year survival of CCA patients is very low, at approximately 18–35.5%, even after curative resection [5,6]. *Opisthorchis viverrini* (OV) infection is a major risk factor of CCA in southeast Asia. Evidence of current or past OV infection can be detected in the serum IgG in CCA patients. This has been reported at high sensitivity [7,8,9]. Several reports suggested that the major mechanism of OV-induced CCA carcinosis is the generation of oxidative stress which initiates DNA mutation as well as genomic instability and alteration [7,8,9,10,11,12,13,14]. The accumulation of mutations leads to the transformation of a normal bile duct into timorous tissue.

DNA mismatch repair protein (MMR) is recognized as the critical DNA repair pathway to repair DNA-base mismatches, insertions and deletions during DNA replication. The MMR proteins are composed of four major proteins, MLH1, MSH2, MSH6 and PMS2 [15]. Many reports suggest that a deficiency in the MMR system (dMMR) results in the accumulation of genomic abnormality and can be seen as microsatellites instability (MSI). The locations of MSI could be found in both coding and non-coding sequences, and dMMR/MSI tumors are associated with a high mutation burden, especially cancer-related genes, that could induce carcinogenesis [16]. dMMR is generalized in different cancer types, occurring with different frequencies and signatures: 15–19% in colorectal [17,18,19], 22% in gastric [20] and 16% in liver cancers [21]. Therefore, the detection of MSI can be performed by directly examining the mutations on microsatellite regions or determining the presence of abnormal of MMR proteins [22,23,24,25,26,27,28]. Evaluating MMR protein expression is commonly carried out by a simple screening method. This can predict MSI by determining a deficiency of MMR protein; especially negative expression which has been suggested to correlate with MSI. Cheah PL et al. investigated MMR expression by immunohistochemistry. They showed that dMMR correlated with MSI in colorectal cancer [23]. This finding is also supported by Agostini and team [28]. Currently, dMMR has clinical benefits which can predict a patient’s outcome and response to treatments, such as surgery, chemotherapy, and immunotherapeutic treatments [24,29,30,31,32,33,34]. dMMR has been reported at a low frequency of approximately 2% in non-OV-associated CCA patients [35]. There are also reports showing an effective response to treatments in non-OV-associated CCA patients with dMMR [35,36,37]. Nevertheless, in OV-associated CCA, the prevalence of dMMR and benefits on patient’s survival are still unclear as only limited evidence is available. Therefore, this study aims to evaluate the prevalence of dMMR in CCA patients and determine the correlation between MMR status with the patient’s character and OV infection. Finally, we aim to investigate the prognostic prediction of MMR on the survival outcome of CCA patients.

## 2. Materials and Methods

### 2.1. Patients Sampling and Sample Collection

A total of 102 patients were diagnosed with CCA between January 2005 and December 2017 at Srinagarind Hospital, Khon Kaen University, and were included in this study. Before the samples were collected, informed and written consent was obtained from all patients. After surgery, tissue and serum were kept by the Cholangiocarcinoma Research Institute (CARI), Khon Kaen University. The clinicopathological information including gender, age, tumor location, histology type, stage (primary tumor; T, lymph node; N, or distant metastasis; M) and the survival of patients (time started at the surgery date until death or last day of follow-up) were provide by CARI. For exclusion criteria, the patients who received palliative surgery, carcinoma in situ or inadequate tissue and serum samples were excluded from the study. The study protocol was performed based on the Declaration of Helsinki and was approved by the Human Research Ethics Committee, Khon Kaen University (HE641363).

### 2.2. Antibodies

Antibodies against MMR proteins performed in this study were the following: rabbit monoclonal anti-MLH1 antibody (dilution 1:100, #ab92312) (Abcam, Cambridge, MA, USA), mouse monoclonal anti-MSH2 antibody (dilution 1:50, #33-7900) (Invitrogen, Carlsbad, CA, USA), rabbit monoclonal anti-MHS6 antibody (dilution 1:100, #ab92471) (Abcam, Cambridge, MA, USA) and anti-PMS2 antibody (dilution 1:25, #ab110638) (Abcam, Cambridge, MA, USA).

### 2.3. CCA Tissue Microarray (TMA)

CCA tissues with matched sera were used for TMA preparation. Normal liver tissue was used as a control. A 2.0 mm diameter needle tissue microarrayer was used to prepare the TMA block. Each block was produced with 70 cores, and 4 µm thick sections were prepared by cutting the TMA block and mounting on coated glass slides.

### 2.4. Immunohistochemistry (IHC)

TMAs were performed to detect MMR proteins using the IHC technique. The TMA samples were deparaffinized and rehydrated in xylene and ethanol (100%, 90%, 80% and 70%), respectively. Retrieval was then preformed in 1xTris-EDTA using a pressure cooker (MLH1 and MSH6) and autoclave (MSH2 and PMS2). Endogenous peroxidase activity and non-specific binding were performed using 0.3% (v/v) of hydrogen peroxide (H_2_O_2_) and 10% skim milk, for 10 min for each step. Primary antibodies were then added and incubated at 4 °C overnight. After washing with phosphate buffer saline (PBS) plus 0.1% Tween 20, horseradish peroxidase (HRP)-conjugated secondary antibodies were added and incubated for 1 h. After washing, a 3,3′ diaminobenzidine tetrahydrochloride (DAB) substrate kit (Vector Laboratories, Inc., CA) was used to develop the signal. Mayer’s hematoxylin was used for the counterstain. Dehydration was performed with 70%, 80%, 90% and 100% ethanol and xylene, respectively. After mounting, tissue sections were observed under a light microscope. The status of MMR proteins was evaluated by pathologists when the positive control exhibited a signal after staining. The positive control was composed of internal and external positive controls. Internal positive controls included adjacent normal liver tissues, stomal cells and lymphocytes, while the external control was normal liver tissue.

### 2.5. Evaluation of MMR Protein Expression

The nuclear immunoreactivity of the four markers was examined by evaluating the response in tumor cells. Hepatocytes and lymphocytes in the sections served as the positive controls. Tumor cells in the areas of interest were categorized as positive if their immunoreactive intensity was equal to or greater than that of the positive controls for each marker. Conversely, if the tumor cells exhibited a complete lack of immunoreactivity, they were deemed negative for each marker. The evaluation of tissue sections was conducted independently by two observers who were blinded to the clinical data. In the case of any discrepancies, a multi-viewer microscope was consulted to reach a consensus. For IHC staining, deficient MMR (dMMR) was defined as at least 1 protein showing loss of expression [38]. No protein loss of expression was defined as adequate MMR (pMMR).

### 2.6. Detection of Immunoglobulin G (IgG) Antibodies against Opisthorchis viverrini (OV) Using Enzyme-Linked Immunosorbent Assay (ELISA)

Indirect ELISA was used to investigate the level of immunoglobulin G (IgG) antibodies against OV antigen. Ninety-six-well plates were coated with OV antigen at a concentration of 1500 µg/mL in 1x phosphate buffer saline (PBS), at pH 7.4, and then incubated at 4 °C overnight. Plates were washed with 1x PBS with 0.05% Tween 20 and then blocked with 3% skim milk in 1x PBS (250 μL/well) at 37 °C for 1 h. Patient serum dilution at 1:6000 in 3% skim milk was added at 100 μL/well and all samples were duplicated. Then, the plates were incubated at 4 °C overnight. After washing, horseradish peroxidase (HRP)-conjugated goat anti-human IgG, at a dilution of 1:3000 in 3% skim milk, was added at 100 μL/well. Next, the plates were incubated at 37 °C for 2 h. After washing, 100 μL/well orthophenylene diamine hydrochloride (OPD) (Zymed, CA, USA) was added and incubated for 30 min in order to develop the signal. The reaction was stopped using 4N sulfuric acid (H_2_SO_4_) and the optical density (OD) was measured using a microplate reader (at 492 nm) (Tecan, Austria). Patients with positive detection with arbitrary units (AU) of OV IgG above percentile 75 were defined as having a high level of OV IgG.

### 2.7. Statistical Analysis

Statistical analysis was carried out using the Statistical Package for the Social Sciences (SPSS) software v.25. Categorical data are presented as percentage and continuous data as mean with SD or median with range. Comparisons between MMR status with patient’s characteristics and OV IgG level were carried out using the chi-square test or Fisher’s exact test, as appropriate. Survival analysis was performed using the Kaplan–Meier method and compared by log-rank test. Univariate and multivariate analyses were performed using the Cox regression model to identify prognostic factors. A *p*-value less than 0.05 was considered statistically significant.

## 3. Results

### 3.1. Prevalence of MMR Status in CCA Patients

A total of 102 CCA patients were evaluated for MMR, i.e., for four of the most common MMR proteins MLH1, MSH2, MSH6 and PMS2, using IHC. Positive expression was indicated by IHC staining if their immunoreactive intensity was equal to or greater than that of the positive controls, while negative expression showed no staining signal (Figure 1 and Figure 2). Among the 102 cases, we found a loss of MLH1, MSH2, MSH6 and PMS2 in 23 cases (22.5%), 4 cases (3.9%), 1 case (0.9%) and 0 cases (0%), respectively. dMMR identification included 23 cases (22.5%) for dMMR and 79 cases (77.5%) for pMMR (Table 1). In addition, dMMR group revealed 52.2% (*n* = 12/23) as single protein deficiency (MLH1), 13.0% (*n* = 3/23) as double protein deficiency (MLH1 + MSH2) and 4.3% (*n* = 1/23) as triple protein deficiency (MLH1 + MSH2 + MSH6).

### 3.2. Patient Characteristics and MMR Status in CCA

The total of 102 CCA samples included 69 (61.1%) from males and 44 (38.9%) from females, while the median age was 60 years (range 39–79). The anatomical location of the tumor comprised 77 (75.5%) intrahepatic CCA and 25 (25.5%) extrahepatic CCA. Of these, 53 (52%) had a microscopically positive surgical margin. For the histological type, 44 (43.1%) were papillary. Lymph node metastases were found in 69 (67.6%). Among the 102 patients, there were 79 (79%) presenting with a TNM late stage (III-IV). A total of 29 (28.4%) patients received adjuvant chemotherapy (CMT). Most patients, 72 (70.6%), presented with a positive OV IgG antibody status. A comparison of patient’s characteristics and OV IgG status with MMR status showed no statistically significant differences in this study (Table 2).

### 3.3. The Correlation of Patient Survival with Their Clinical Features, Level of Serum OV IgG and MMR Status

The median survival time (MST) of all patients was 13 months (mo.). Univariate analysis showed five factors that correlated significantly with overall survival (OS) of the CCA patients: (1) patients who were diagnosed as positive for surgical margin had a significantly shorter survival than those with a negative margin (MST = 12.4 vs. 18.1 mo., HR = 1.83 (95% CI: 1.22–2.74), *p* = 0.003); (2) positive lymph node metastasis showed a markedly lower survival time than negative status (MST = 12.5 vs. 18.8 mo., HR = 1.78 (95% CI: 1.15–2.75), *p* = 0.009); (3) patients with late-stage CCA (II-IV) showed a significantly shorter survival than those with early-stage (I-II) disease (MST= 12.5 vs. 19.2 mo., HR = 1.93 (95% CI: 1.19–3.13), *p* = 0.006); (4) surgical patients who received adjuvant chemotherapy had a longer survival than patients who received only surgical treatment (MST= 17.4 vs. 11.7 mo., HR 0.61 (95% CI: 0.39–0.96), *p* = 0.031); and (5) CCA patients presenting with dMMR had a significantly better survival time than CCA patients with pMMR (MST = 17.0 vs. 12.9 mo., HR = 0.50 (95% CI: 0.30–0.85), *p* = 0.008) (Table 3 and Figure 3). In contrast, no statistically significant difference was found in age, gender, tumor location, histological type or OV IgG antibody status.

Subsequently, the significant factors in univariate analysis were further analyzed by multivariate analysis to identify prognostic factors using the Cox regression model. Results showed that surgical margin, adjuvant chemotherapy, and MMR status were independent factors for the prediction of the outcome of CCA patients (HR = 1.61 (95% CI: 1.01–2.58), 0.62 (95% CI: 0.39–0.98) and 0.81 (95% CI: 0.34–0.97), *p* = 0.049, 0.041 and 0.41, respectively) (Table 3).

### 3.4. Subgroup Analysis on Survival of TNM Stage by MMR Status

Since the tumor stage had a significant effect on the patient’s survival, it may affect other factors relating to survival analysis. dMMR proteins are potential prognostic factors for clinical benefits in the early stage of colorectal cancer. Thus, survival analysis was further explored according to early and late tumor staging.

In early-stage CCA, comprising stages I and II, a total of 23 patients were dived into 2 groups consisting of 4 (17%) dMMR and 19 (83%) pMMR. CCA patients in the early-stage of the disease who had dMMR status had a significantly better survival than pMMR status (MST = 40.6 vs. 18.8 mo., HR = 0.18 (95% CI: 0.04–0.82), *p* = 0.027) (Figure 4A). For late-stage CCA (III-IV), there was no statistically significant difference found between dMMR and pMMR groups (MST = 15.9 vs. 11.5 mo., HR = 0.61 (95% CI: 0.35–1.05), *p* = 0.076).

### 3.5. Subgroup Analysis on Survival of CCA with Adjuvant Chemotherapy by MMR Status

Our aim was to evaluate the chemotherapeutic (CMT) effects on CCA patients who showed dMMR. Several reports suggest that dMMR can be applied with clinical benefit for treatments such as chemotherapy and immunotherapy in several cancers, especially colorectal cancer [17,18,19,20,21]. All CCA patients received surgical resection and 29.9% of patients received adjuvant chemotherapy. Therefore, surgical patients were separated into those receiving CMT and those not receiving CMT. dMMR patients who received CMT had significantly better survival rates than patients who received surgery alone (MST = 23 vs. 13 mo., HR = 0.28 (95% CI: 0.10–0.81), *p* = 0.012) (Figure 4B), while there was no statistically significant difference on survival in pMMR patients who received CMT and surgery alone (MST = 15.6 vs. 11.7 mo., HR = 0.87 (95% CI: 0.52–1.44), *p* = 0.581) (Figure 4C).

## 4. Discussion

Cholangiocarcinoma has its highest incidence worldwide in Thailand. Its major risk factor is *Opisthorchis viverrini* (OV) infection. This information has been verified in several studies which applied different methods to determine OV infection using faecal, urine and serum specimens [7,9,12]. The evidence of current or past OV infection can be tested by detecting serum IgG in patients which arises after OV infection: the ELISA technique has high sensitivity (99.2%) and specificity (93%) [9]. This technique was validated by Titapun A et al. who showed that the serum of 73% of CCA patients was positive for OV IgG [7]. Similarly, we also found that 70.6% of CCA patients were positive for IgG for OV infection. This finding suggested that about three quarters of CCA patients had an OV infection [7]. This information indicated that the cause of CCA in our cohort was associated with OV infection.

Several mechanisms initiated by OV infection induce normal bile ducts to transform into cancer. The most of important mechanism is inducing oxidative stress to damage biomolecules such as lipids, proteins and DNA, which leads to disfunction and mutation [39,40,41,42,43]. The most commonly initiated process of carcinogenesis is genomic mutation in tumor-related genes such as oncogenes, tumor suppressor genes and DNA repair genes, which lead to altered DNA sequences, i.e., genetic instability. Genetic instability is composed of two different forms, chromosomal instability (CIN) and microsatellite instability (MSI). For CIN the mechanism usually begins with a high frequency of allelic losses and cytogenetic abnormalities during the cell cycle. While MSI often occurs in DNA replication error by inactivation of genes which are responsible for DNA nucleotides, also known as mismatch repair (MMR). The MMR deficiency leads to DNA replication errors with accumulated mutations in DNA sequences at the nucleotide level. Genomic mutations or alterations are major causes of carcinogenesis and tumor progression. Several cancers have different frequencies and prevalence of dMMR, which range from 15 to 22% [17,18,19,20,21]. In CCA, Le TD et al. showed the prevalence of dMMR in approximately 2% of non-OV-associated CCA cases [35]. In the present study, dMMR prevalence in OV-associated CCA was approximately 22.5%, 10-fold higher than previous reports in non-OV-associated CCA. In addition, we also revealed that dMMR consisted of a high proportion of negative MLH (22.5%), while there was a low proportion in MSH2 (3.9%) and MHS6 (0.9%) as well as no negative cases found in PMS2. The high proportion of the absence MLH1 might be explained as a sporadic dMMR event where MLH1 usually silences expression via epigenetic alteration, especially hypermethylation on the MLH1 promoter at the CpG island. This epigenetic mechanism could suppress MLH1 expression in dMMR as reported in several studies [44,45,46,47,48]. Those reports are concordant that MLH1 is usually impacted by many oncogenic initiators that might be direct or indirect in action. The prevalence of dMMR in our study was as high as that detected in colorectal and gastric cancer [17,18,19,20].

To explain the relationship between OV infection and dMMR in CCA, our results showed that a total of 73.9% of CCA patients with dMMR status were detected in individuals with a positive OV IgG status. Therefore, OV infection might cause a deficiency in MMR proteins via an oxidative stress cascade resulting in hypermethylation of a promotor of these proteins. In the northeast of Thailand, OV infection is the cause of oxidative stress production that leads to the generation of DNA mutations, as has been reported in several studies [11,12,13]. A previous study by Loilome et al., revealed that oxidative stress correlated significantly with MSI in OV-associated CCA. Their study investigated the association between oxidative stress and MSI level in OV-associated CCA. They found that antioxidant enzymes including SOD2 and CAT were decreased, while DNA repair enzymes were imbalanced (i.e., were both over- and under-expressed when compared to non-tumor tissue), and this abnormality generated MSI. In addition, their study also showed that MSI occurred in all patients in their study composing MSI-high as 69% and MSI-low as 31% [43]. Therefore, their study indicated that oxidative stress causes a DNA repair imbalance resulting in MSI in OV-associated CCA. However, there are gaps in Loilome’s study; for instance, the status of OV infection and MMR protein were not determined. Therefore, our investigation aimed to determine the status of OV infection by detecting IgG antibody and MMR protein expression, and to investigate whether there was a correlation between both factors. Thus, the present study examined OV IgG antibodies using an ELISA assay which is the major tool for detecting OV infection [7,8,9], while MMR protein expression of MLH1, MSH2, MHS6 and PMS2, was determined using an IHC assay [49,50,51]. Our findings suggested that three quarters of CCA patients in our cohort had a positive status for the OV IgG antibody. In addition, 73.9% of CCA patients who had positive OV IgG antibody status, presented a dMMR status. Although our study showed no significant difference between the prevalence of OV infection and dMMR status (*p* = 0.691), this finding is informative concerning dMMR and could be a good prognostic factor in the survival of CCA patients, especially in early-stage patients and patients who received chemotherapy, and not only OV-associated CCA but also in non-OV-associated CCA patients.

The retrospective study of the roles of MMR status in survival and predicting the outcome after treatments was revealed for OV-associated CCA. Our findings showed that surgical patients who had dMMR status were significantly, positively correlated with longer survival than those with pMMR status. In addition, MMR status was also an independent factor for the prognostic prediction of the outcome of CCA patients who underwent resection, as found in several other cancers [24,29,30,36]. In addition, several studies suggest that dMMR status is a potential factor for predicting a good outcome in cancer patients with early-stage disease [52,53,54,55]. Jin Z and Sinicrope AF suggested that dMMR/MSI-H in non-metastatic colorectal cancer (CRC) patients showed a good outcome after treatment by improving disease-free survival (DFS) and overall survival in the early stage [52]. This information is supported by Kawakami et al. who compared CRC patients who had an MMR-deficient/MSI-H status between stages II and III and compared the outcome of dMMR and pMMR in the same stage. The result showed that MMR-deficient/MSI-H status in stage II had a better outcome after treatment than stage III, and this outcome was also better than in patients with pMMR in the same stage (Figure 3). In the present study, we separated the CCA patients into early and late stages and examined the role of dMMR on the survival outcome of CCA patients in both stages. Interestingly, we found that patients with dMMR status had longer survival than those with pMMR status by approximately 2-fold. In late-stage patients with dMMR, the trend to better survival was also found, although without statistical significance. Moreover, we also demonstrated the importance of dMMR on clinical benefits by comparison of surgery alone and surgery with adjuvant CMT. Results from subgroup analysis of dMMR status showed that CCA patients who received adjuvant CMT had significantly longer survival than CCA patients who received only surgery. Our result was similar to that found in several cancers that showed the role of dMMR on improvement of treatment efficiency in clinical benefits by combining resection with either chemotherapy or immunotherapy compared with resection alone [24,29,30,31,32,33,34,36,37,56]. This finding may be useful for the prognosis of a patient’s outcome and treatment plan in OV-associated CCA.

There are limitations of this study: (1) The determination of MSI status was not performed to investigate the mutation of patients with MMR deficiency. (2) The number of patients in early-stage disease, especially in dMMR, were small and an increased number of patients should be provided to validate the results.

## 5. Conclusions

Therefore, the present study suggests a high prevalence of dMMR status in OV-associated CCA and this is correlated with a good survival outcome. It is an independent prognostic factor in early-stage CCA. Moreover, dMMR can be applied in clinical usage to predict the response of adjuvant CMT in CCA patients after curative resection.

## Figures and Tables

**Figure 1 cancers-15-04831-f001:**
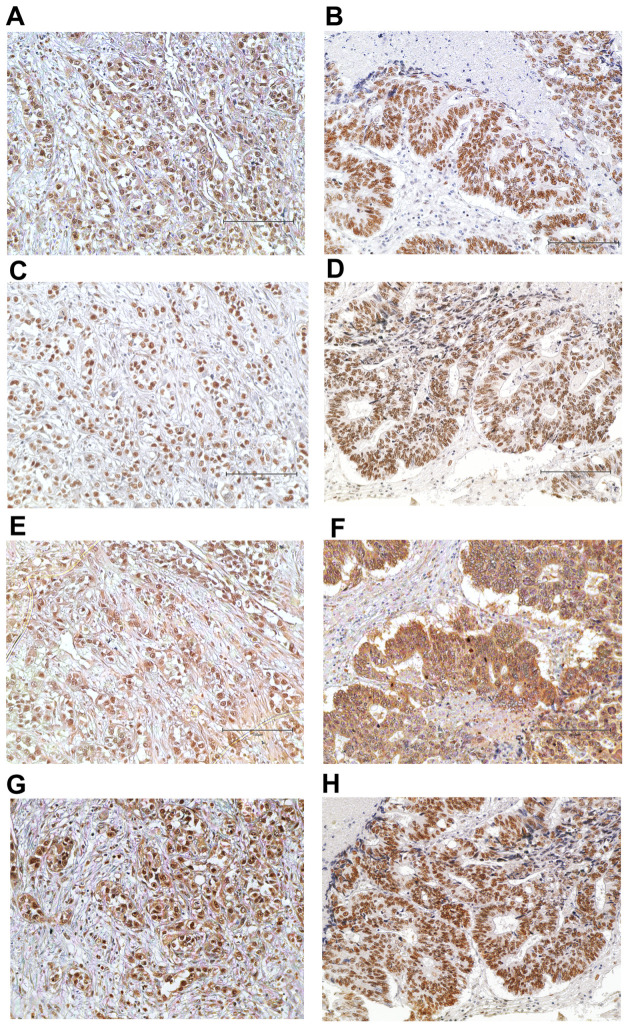
The representative figures of pMMR status. Positive for MLH1 (**A**), MSH2 (**C**), MSH6 (**E**) and PMS2 (**G**) in the same case. Positive for MLH1 (**B**), MSH2 (**D**), MSH6 (**F**) and PMS2 (**H**) in the same case. Immunohistochemical staining; original magnification ×200; scale bar, 150 μm.

**Figure 2 cancers-15-04831-f002:**
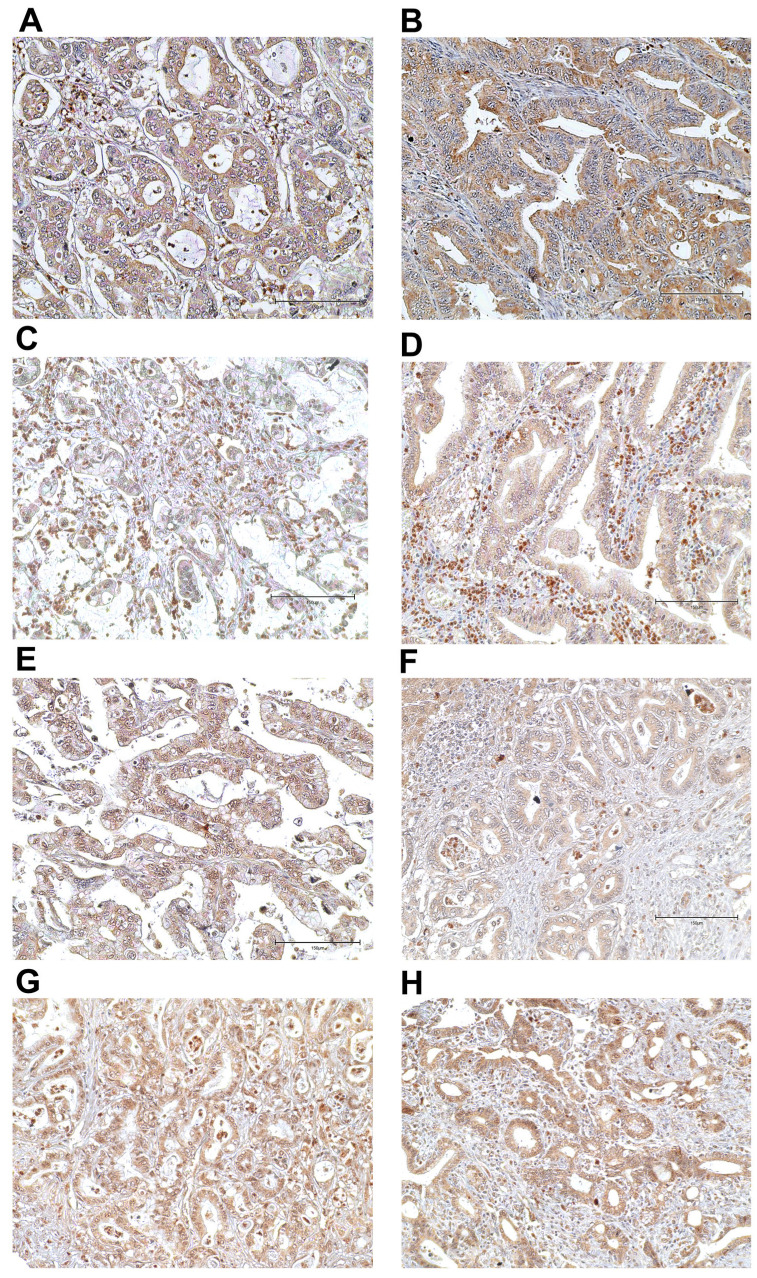
The representative figures of dMMR status. Negative for MLH1 (**A**) and MSH2 (**C**) but positive for MSH6 (**E**) and PMS2 (**G**) in the same case. Negative for MLH1 (**B**), MSH2 (**D**) and MSH6 (**F**) but positive for PMS2 (**H**) in the same case. Immunohistochemical staining; original magnification ×200; scale bar, 150 μm.

**Figure 3 cancers-15-04831-f003:**
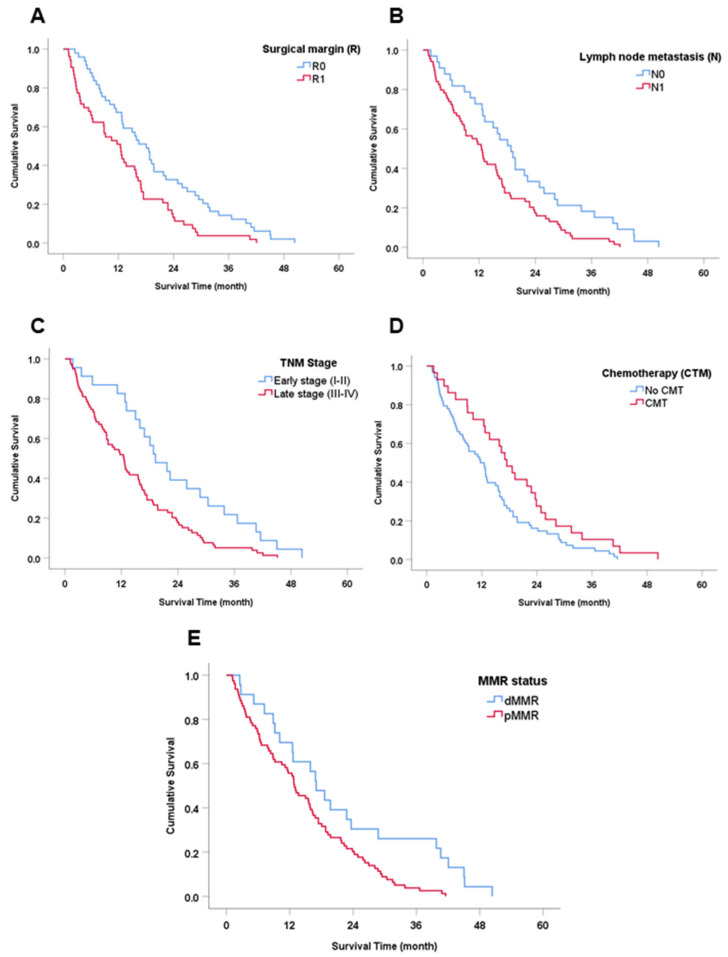
Survival (Kaplan–Meier) analysis of clinicopathological features and MMR proteins. (**A**) Surgical margin (R). (**B**) Lymph node metastasis. (**C**) TNM stage. (**D**) Chemotherapy treatment (CMT). (**E**) MMR status.

**Figure 4 cancers-15-04831-f004:**
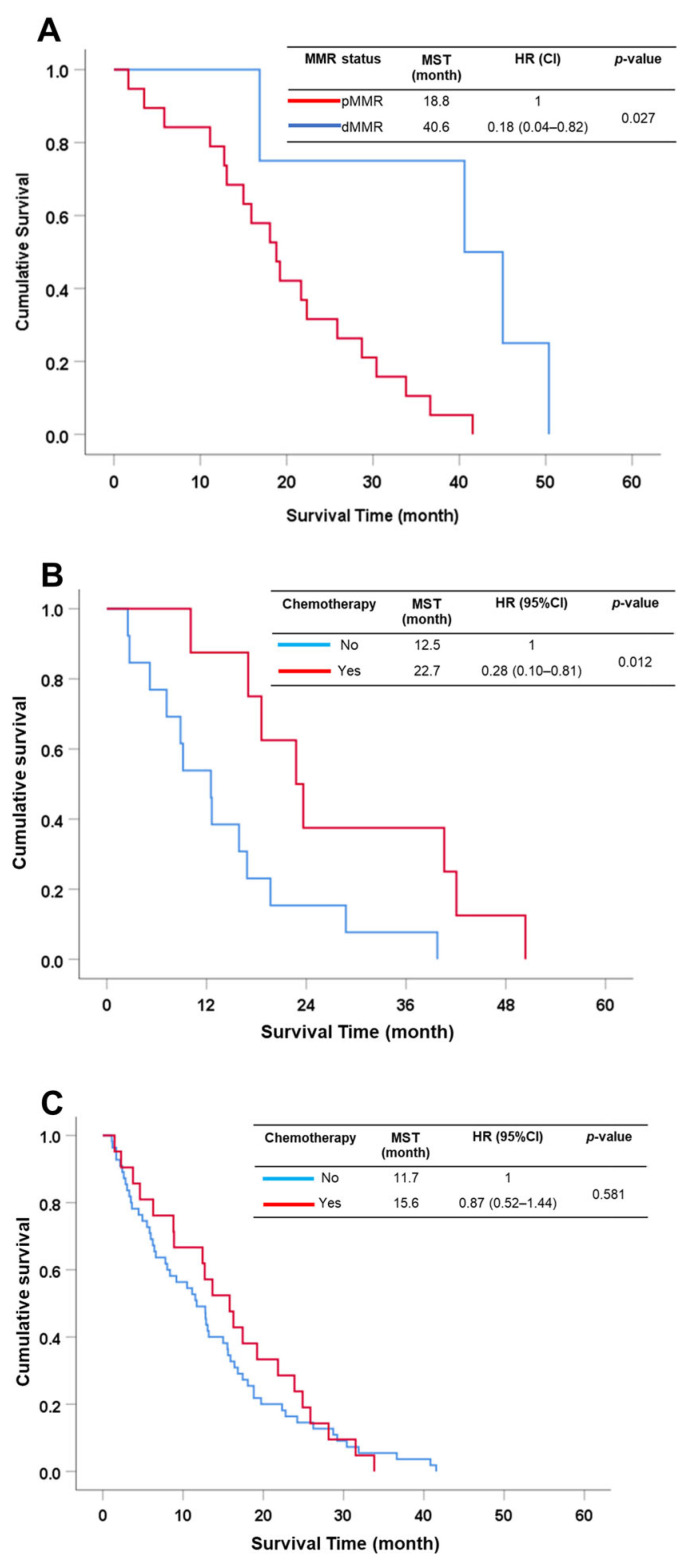
Survival analysis of subgroup analysis. (**A**) MMR status in the early stage of CCA patients. (**B**) CCA patients with dMMR status with or without chemotherapy treatment. (**C**) CCA patients with pMMR status with or without chemotherapy treatment.

**Table 1 cancers-15-04831-t001:** The MMR status of CCA patients.

MMR Protein	IHC Status		Total
	Negative	Positive	
MLH1	23 (22.5%)	79 (77.5%)	102 (100%)
MSH2	4 (3.9%)	98 (96.1%)	102 (100%)
MSH6	1 (0.9%)	101 (99.1%)	102 (100%)
PMS2	0 (0%)	102 (100%)	102 (100%)
**MMR status**	**dMMR**	**pMMR**	102 (100%)
	23 (22.5%)	79 (77.5%)	

MMR = DNA mismatch repair protein; dMMR = deficient MMR; pMMR = proficient MMR.

**Table 2 cancers-15-04831-t002:** Patient’s characteristics and MMR protein status in CCA.

Features	n (102, 100%)	MMR Status		*p*-Value
		dMMR N (23, 22.5%)	pMMR N (79, 77.5%)	
Age (yrs.)				
<60	50 (49%)	9 (39.1%)	41 (51.9%)	0.281
≥60	52 (51%)	14 (60.9%)	38 (48.1%)	
Gender				
Male	61 (59.8%)	14 (60.9%)	47 (59.5%)	1.000 ^#^
Female	41 (40.2%)	9 (39.1%)	32 (40.5%)	
Tumor location				
Intrahepatic CCA	77 (75.5%)	21 (91.3%)	56 (70.9%)	0.055 ^#^
Extrahepatic CCA	25 (24.5%)	2 (8.7%)	23 (29.1%)	
Surgical margin				
Negative	49 (48%)	10 (43.5%)	39 (49.4%)	0.644 ^#^
Positive	53 (52%)	13 (56.5%)	40 (50.6%)	
Histological type				
Papillary	44 (43.1%)	10 (43.5%)	34 (43%)	1.000 ^#^
Others	58 (56.9%)	13 (56.5%)	45 (57%)	
Lymph node metastasis				
No	33 (32.4%)	8 (34.8%)	25 (31.6%)	0.803
Yes	69 (67.6%)	15 (65.2%)	54 (68.4%)	
TNM stage				
I–II	23 (23%)	4 (17.4%)	19 (24.1%)	0.501
III–IV	79 (79%)	19 (82.6%)	60 (75.9%)	
Adjuvant CMT				
No	68 (70.1%)	13 (61.9%)	55 (72.4%)	0.354
Yes	29 (29.9%)	8 (38.1%)	21 (27.6%)	
OV antibody				
Negative	30 (29.4%)	6 (26.1%)	24 (30.4%)	0.691
Positive	72 (70.6%)	17 (73.9%)	55 (69.6%)	

^#^ Fisher’s exact test.

**Table 3 cancers-15-04831-t003:** The survival analysis of clinicopathological features and MMR protein expression.

Patient’s Characteristics	n (102)	MST [Months (95% CI)]	Crude HR (95% CI)	*p*-Value	Adjusted HR (95% CI)	*p*-Value
Gender						
Male	61	15.9 (11.7–20.1)	1	0.695		
Female	41	11.7 (7.2–16.2)	1.08 (0.73–1.62)			
Age (yrs.)						
<60	50	12.8 (8.9–16.6)	1	0.403		
≥60	52	15.5 (10.7–20.3)	0.85 (0.57–1.24)			
Tumor location						
Intrahepatic CCA	77	12.8 (9.1–16.5)	1	0.805		
Extrahepatic CCA	25	17 (14.0–19.9)	0.94 (0.60–1.50)			
Margin						
Negative	49	18.1 (14.4–21.7)	1	0.003 *	1	0.049 *
Positive	53	12.4 (8.4–16.4)	1.83 (1.22–2.74)		1.60 (1.01–2.58)	
Histological type						
Papillary	44	16.9 (11.2–22.6)	1	0.285		
Others	58	12.5 (8.7–16.4)	1.24 (0.84–1.84)			
Lymph node metastasis (N)						
No	33	18.8 (14.5–23.1)	1	0.009 *	1	0.752
Yes	69	12.5 (9.8–15.3)	1.78 (1.15–2.75)		0.91 (0.51–1.62)	
TNM stage						
I–II	23	19.2 (13.6–24.9)	1	0.006 *	1	0.085
III–IV	79	12.5 (9.9–15.2)	1.93 (1.19–3.13)		1.81 (0.92–3.56)	
Adjuvant CMT						
No	68	11.7 (8.0–15.4)	1	0.031 *	1	0.041 *
Yes	29	17.4 (13.3–21.5)	0.61 (0.39–0.96)		0.62 (0.39–0.98)	
OV antibody						
Negative	30	12.8 (7.2–18.4)	1	0.164		
Positive	72	13.7 (10.4–16.9)	1.37 (0.88–2.12)			
MMR protein status						
pMMR	79	12.9 (9.7–16.1)	1	0.008 *	1	0.041 *
dMMR	23	17.0 (12.8–21.2)	0.50 (0.30–0.85)		0.58 (0.34–0.97)	

*, Statically significant; MST, median survival time.

## Data Availability

All materials are available upon request.

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
