# Peer review of "Mismatch Repair Deficiency Is a Prognostic Factor Predicting Good Survival of Opisthorchis viverrini-Associated Cholangiocarcinoma at Early Cancer Stage"

_cancers, 2023, doi:10.3390/cancers15194831_

Round 1

Reviewer 1 Report

I have evaluated the work of Natcha Khuntikeo et al., which aims to demonstrate the prevalence of dMMR (deficient mismatch repair) in cholangiocarcinoma (CCA) and its potential correlations with patient characteristics, OV (Opisthorchis viverrini) infection, and prognosis. Unfortunately, I believe that the study has significant gaps that require further work. I summarize why I consider it immature for publication or unsuitable for a high-impact factor journal. There is an excessive emphasis on a nearly clear, precise correspondence between dMMR and MSI (microsatellite instability), which is not entirely accurate. Also, the statement "The accumulation of MSI leads to the initiation of carcinogenesis" is not precise. While dMMR is a good predictor of the MSI phenotype, there isn't a mechanistic correspondence. In section 2.4, regarding the immunohistochemistry methods, I would have expected not only a positive control (which is challenging) but also a negative control for the experiments (perhaps by substituting the primary antibody with an irrelevant protein or a non-human protein of the same subclass). The authors correctly mention that dMMR is evaluated by assessing MLH1, MSH2, MSH6, and PMS2 (see, for example, PMID: 34113169), but nowhere in their manuscript do they evaluate PMS2. This represents a significant gap in their work. In line 281, they state "...MSI generation (the product of dMMR)..." – this statement is not scientifically accurate, as the correspondence between dMMR and MSI is not absolute or direct. The entire study is built on a biological correlation between OV infection, oxidative stress, and MSI status, yet this correlation is not found in the data. In line 304, the authors provide explanations that are overly complex and hypothetical, reiterating in lines 312-315 that OV infection might influence dMMR status through oxidative stress. The discussion is convoluted and extravagant, lacking support from the data. The comparison with colorectal cancer on dMMR is unsupported, as these are biologically distinct diseases.

The English quality is not good and requires extensive revision.

Author Response

  1. There is an excessive emphasis on a nearly clear, precise correspondence between dMMR and MSI (microsatellite instability), which is not entirely accurate. Also, the statement "The accumulation of MSI leads to the initiation of carcinogenesis" is not precise. While dMMR is a good predictor of the MSI phenotype, there isn't a mechanistic correspondence.

Respond to reviewer: thank you reviewer for the suggestion, we agree with this comment. as such we have rewritten the information to reduce excessive emphasis between dMMR and MSI in line 60-63 and highlight in yellow color.

  1. In section 2.4, regarding the immunohistochemistry methods, I would have expected not only a positive control (which is challenging) but also a negative control for the experiments (perhaps by substituting the primary antibody with an irrelevant protein or a non-human protein of the same subclass). 

Respond to reviewer: thank you reviewer for the suggestion. We agree with this comment. The description of positive control was informed in section 2.4 line 122-126. The positive control composed internal and external positive controls. The internal positive controls included adjacent normal liver tissues, stomal cells and lymphocytes, while external control was normal liver tissue.

For the negative control, we did not validate the negative control, since the antibodies used in this study are commercial antibodies. The validation of specificity is performed by those companies.

  1. The authors correctly mention that dMMR is evaluated by assessing MLH1, MSH2, MSH6, and PMS2 (see, for example, PMID: 34113169), but nowhere in their manuscript do they evaluate PMS2. This represents a significant gap in their work.

Respond to reviewer: thank you very much reviewer for suggestion, we agree with this comment.  we detected PMS2 using IHC and found that PMS2 had positive staining in all cases in this study as shown in Table 1 and Figure 1-2.

  1. In line 281, they state "...MSI generation (the product of dMMR)..." – this statement is not scientifically accurate, as the correspondence between dMMR and MSI is not absolute or direct. 

Respond to reviewer: thank you very much reviewer for suggestion, we agree with this comment, and we rewritten in line 295-299 and highlight in yellow color.

  1. The entire study is built on a biological correlation between OV infection, oxidative stress, and MSI status, yet this correlation is not found in the data. In line 304, the authors provide explanations that are overly complex and hypothetical, reiterating in lines 312-315 that OV infection might influence dMMR status through oxidative stress. The discussion is convoluted and extravagant, lacking support from the data. The comparison with colorectal cancer on dMMR is unsupported, as these are biologically distinct diseases.

Respond to reviewer: We agree with this comment. We deleted the overly complex and hypothetical, reiterating in lines 312-315, and have rewritten as in line 311-318, highlighted in yellow color.

Comments on the Quality of English Language

The English quality is not good and requires extensive revision.

Respond to reviewer: We agree with this comment. The manuscript was improved the English quality by Professor Trevor N. Petney, the Publication Clinic at Khon Kaen University, Thailand. All changes in our manuscript are tagged by tagged change and highlight in yellow color.

Reviewer 2 Report

The authors present a very nice paper on MMR and cholangiocarcinoma, in this case with a peculiar aetiology, related to their geographic location.

In mM&M: "Retrieval was then performed in 1X Tris-EDTA using a pressure 108 cooker (MLH1 and MSH6) and autoclave (MSH2)" - how about PMS2?

This is a major limitation nowadays! One cannot have this type of study using only 3 proteins, especially when they has a high loss of MLH1

The remaining of the study is ok, but I would also like to see more emphasis in the discussion of the high rate of loss of MLH1; this biomarker is highly sensitive to the pre-analytical stage and may have false negatives; the conjugation of PMS2 is therefore mandatory.

I recommend the authors to do the PMS2 and resubmit the manuscript with new data analysis. If they do not have the antibody I can recommend some labs is Portugal that may be of help

The article needs some English revision. Some statements sound like bullet points and there are missing words that would connect sentences and provide a more comprehensible reading.

Author Response

Comments and Suggestions for Authors

The authors present a very nice paper on MMR and cholangiocarcinoma, in this case with a peculiar aetiology, related to their geographic location.

  1. In mM&M: "Retrieval was then performed in 1X Tris-EDTA using a pressure 108 cooker (MLH1 and MSH6) and autoclave (MSH2)" - how about PMS2?

Respond to reviewer: Regarding the antigen retrieval process, we attempted to perform the same conditions to retrieve antigens in tissues using a pressure cooker. However, there are only two proteins MLH1 and MSH6 that were detected by IHC, while MSH2 and PMS2 could not be detected under these conditions. Therefore, we optimized this condition and, we found that retrieval antigens under autoclave enabled detection of MSH2 and PMS2.

  1. This is a major limitation nowadays! One cannot have this type of study using only 3 proteins, especially when they have a high loss of MLH1

Respond to reviewer: thank you for the suggestion, we agree with this comment. we performed IHC to detect PMS2 expression and we found that PMS2 had positive stanning in all CCA patients (n=102) as shown in Table 1. Thus, this study showed a high loss of MLH1 in CCA patients.

  1. The remaining of the study is ok, but I would also like to see more emphasis in the discussion of the high rate of loss of MLH1; this biomarker is highly sensitive to the pre-analytical stage and may have false negatives; the conjugation of PMS2 is therefore mandatory.

Respond to reviewer: We agree with this comment.  We added a discussion of the high rate of loss of MLH1 in line 284-291 and highlight in yellow color. In addition, PMS is detected as positive staining in all cases.

  1. I recommend the authors to do the PMS2 and resubmit the manuscript with new data analysis. If they do not have the antibody I can recommend some labs is Portugal that may be of help

Respond to reviewer: To detect PMS2 we used IHC and found that PMS2 had positive staining in all cases in this study as shown in Table 1 and Figure 1-2.

Comments on the Quality of English Language

The article needs some English revision. Some statements sound like bullet points and there are missing words that would connect sentences and provide a more comprehensible reading.

Respond to reviewer: We agree with this comment. The manuscript was improved the English quality by Professor Trevor N. Petney, the Publication Clinic at Khon Kaen University, Thailand. All changes in our manuscript are tagged by tagged change and highlight in yellow color.

Reviewer 3 Report

In this paper the authors examined the prognostic role of MSI-status (evaluated immunohistochemically) in cholangiocarcinomas. The ppaer is interesting but some points need to be added or modifed, as follows:

1. Abstract - English revision is recommended to avoid improper terms such "serums", "MMR status was identified".....

2. Abstract - it is necessary to mention which IHC markers were used.

3. Introduction - main text - As the paper disccussion is not focused upon Thailand only it is necessary mentioning the incidence all over Asia and Europe (PMID: 23188430)

4, Methods - check again the spelling mistakes (e.g page 3 - row 95 - "antibodies were used..."), These mistakes and lack of the subject is frequent. English quality is mandatory to be checked by a professional team and a Certificate of English correction is necessary to be attached.

5. Study design - page 3 - rows 96-99 - The dMMR status should be checked with four markers (PMID: 31966075): MLH-1/PMS-2 and MSH-2/MSH-6 - It is mandatory to add PMS-2. Otherwise, the study does not have a proper design

6. Methods - page 3 - row 106 - IHC does not detect"the level" of proteins - it is a qualitative stain - please reword

7. Results - row 161 - it is unclear what those 22.5% represents.

8. Results - without PMS-2 the results does not have any value. The cases need to be re-interpreted and statistical assessment re-done after evaluation of PMS-2 expression

9. The paper can be re-evaluated after adding PMS-2 stain and Sanger evaluation for dMMR cases. Otherwise, the paper does not have a scientific base. 

English quality is mandatory to be checked by a professional team and a Certificate of English correction is necessary to be attached.

Author Response

Comments and Suggestions for Authors

In this paper the authors examined the prognostic role of MSI-status (evaluated immunohistochemically) in cholangiocarcinomas. The ppaer is interesting but some points need to be added or modifed, as follows:

  1. Abstract - English revision is recommended to avoid improper terms such "serums", "MMR status was identified".....

Respond to reviewer: Thank you reviewer for the suggestion, we corrected the two words: “serums into serum” and “MMR status was identified” into “The expression of MMR proteins including MLH1, MSH2, MSH6 and PMS2 was evaluated by immunohistochemistry”.

  1. Abstract - it is necessary to mention which IHC markers were used.

Respond to reviewer: We add the detail of IHC markers in this study including MLH1, MSH2, MSH6 and PMS2 as shown in line 26.

  1. Introduction - main text - As the paper disccussion is not focused upon Thailand only it is necessary mentioning the incidence all over Asia and Europe (PMID: 23188430)

Respond to reviewer: Thank you reviewer for suggestion, we agree with this comment.  We added your suggesting information in the introduction at line 43-44 and yellow color.

  1. Methods - check again the spelling mistakes (e.g page 3 - row 95 - "antibodies were used..."), These mistakes and lack of the subject is frequent. English quality is mandatory to be checked by a professional team and a Certificate of English correction is necessary to be attached.

Respond to reviewer: Thank you reviewer for suggestion our spelling mistake, we correct the sentence in line 95 from “antibodies were used..." to “Antibodies of MMR proteins were performed in the study the following:

  1. Study design - page 3 - rows 96-99 - The dMMR status should be checked with four

markers (PMID: 31966075): MLH-1/PMS-2 and MSH-2/MSH-6 - It is mandatory to add PMS-2. Otherwise, the study does not have a proper design

Respond to reviewer: Thank you reviewer for suggestion, we agree with this comment. We included detection of  PMS2 expression in CCA tissue in our study, and we found that PMS2 has positive staining in all cases of CCA as shown in Table 1.

  1. Methods - page 3 - row 106 - IHC does not detect"the level" of proteins - it is a

qualitative stain - please reword

Respond to reviewer: This has been reworded in line 106 from “the level” into “ TMAs were performed to detect MMR proteins using IHC technique” as shown in line 108.

  1. Results - row 161 - it is unclear what those 22.5% represents.

Respond to reviewer: This has been rewritten as suggested in the line 161 which was an unclear sentence into new sentence in line 163-164 “Among 102 cases, we found a loss of MLH1, MSH2, MSH6 and PMS2 in 23 cases (22.5%), 4 cases (3.9%), 1 case (0.9%) and 0 case (0%), respectively.”

  1. interpreted and statistical assessment re-done after evaluation of PMS-2 expression

Respond to reviewer: We included as suggested the detection of PMS2 expression in CCA tissue in our study, and we found that PMS2 has positive staining in all cases of CCA as shown in Table 1.

  1. The paper can be re-evaluated after adding PMS-2 stain and Sanger evaluation for

dMMR cases. Otherwise, the paper does not have a scientific base. 

Respond to reviewer: Thank you for suggestion, we agree with this comment. We have included the detect of PMS2 expression in CCA tissue in our study, and we found that PMS2 has positive staining in all cases of CCA as shown in Table 1. For the Sanger evaluation for dMMR cases, it is a good idea for the further study. Thank you for the advice.

Comments on the Quality of English Language

English quality is mandatory to be checked by a professional team and a Certificate of English correction is necessary to be attached.

Respond to reviewer: We agree with this comment. The manuscript was improved the English quality by Professor Trevor N. Petney, the Publication Clinic at Khon Kaen University, Thailand. All changes in our manuscript are tagged by tagged change and highlight in yellow color.

Round 2

Reviewer 1 Report

The authors have put in a great effort to improve the manuscript. They responded appropriately to my requests, and I am satisfied with it. I congratulate them for their correctness and patience in enhancing the scientific rigor of their work.

Reviewer 3 Report

Improved paper